# What Determines Forest Farmers’ Participation in Afforestation Programs? Empirical Evidence from a Population-Based Census Survey

**DOI:** 10.3390/ijerph17113962

**Published:** 2020-06-03

**Authors:** Tzong-Haw Lee, Brian Lee, Yu-Long Chen, Lih-Chyun Sun, Hung-Hao Chang

**Affiliations:** 1College of Economics, Fujian Agriculture and Forestry University, Fuzhou 350002, China; tzonghawlee@gmail.com; 2Department of Agricultural Economics, National Taiwan University, Taipei 10617, Taiwan; r07627034@ntu.edu.tw; 3Department of Business Administration, Chaoyang University of Technology, Taichung 413310, Taiwan; s10537902@gm.cyut.edu.tw; 4Department of Urban Industrial Management and Marketing, University of Taipei, Taipei 11153, Taiwan

**Keywords:** afforestation program, multinomial logit model, policy simulation, Taiwan

## Abstract

Afforestation programs have become increasingly significant as policymakers attempt to protect the environment and reduce climate change. Although many studies have examined the participation decisions of forest farm households in afforestation programs, these studies fail to consider different types of these policies. This paper fills this knowledge gap by studying the determinants of forest farms participating in two afforestation programs on plain and sloped land in Taiwan. We construct a population-based sample of forest farms drawn from agricultural census surveys in Taiwan and estimate the multinomial logit model. We find that failing to distinguish between afforestation programs may result in misleading findings. Moreover, socio-demographic and farm production characteristics also affect participation decisions. Additional results from a simple simulation exercise show that forest farms are more likely to enroll in afforestation programs on sloped land, possibly due to lower opportunity costs

## 1. Introduction

Afforestation programs, defined as measures that convert abandoned and degraded land into forests by planting trees, have been identified as a central component of climate change mitigation strategies [1]. In addition to reducing greenhouse gasses through carbon sequestration, afforestation programs are particularly attractive to policymakers since they typically operate by directly paying participants to plant tree cover on land. Thus, countries like China and Spain have implemented policies such as the Conversion of Cropland to Forest Program (CCFP) and the State Forestry Resources Law (SFRL) to lower net carbon emissions while simultaneously providing economic support to households [2,3].

As policymakers continue to encourage individuals to enroll in afforestation programs, one crucial question is how the physical characteristics of the property or land can affect the decision to participate. For example, farmers cultivating plots on steep or sloped land may have greater incentives to participate in afforestation programs since these areas tend to have nutritionally deficient or eroded soils [4]. In contrast, farmers cultivating plots on plain land could have fewer economic incentives to participate since these areas are more agriculturally productive.

Given the importance of afforestation on sustainable development, studies on afforestation have examined the factors that can influence enrollment. Firstly, there is a body of literature that focuses on how household and property characteristics affect participation rates. Nguyen et al. [5] and Legesse et al. [6] find that land ownership, privatization, and security increase participation rates in Vietnam and Ethiopia’s afforestation programs. Similarly, Nielsen et al. [7] develop a revealed choice model and show that socio-demographic factors and the property’s physical characteristics, such as the total area of oak scrub, can affect participation in a conservation program for this species in Denmark. Other studies have considered how policy details and schemes can impact enrollment. Kim and Langpap [8] show that incentive payments positively affect participation in afforestation programs in the United States. Similarly, Tadesse et al. [9] find that restrictions on households’ ability to collect charcoal and timber from forests and a lack of incentives can negatively impact participation rates. Additional work has examined the importance of non-market values and environmental motivations on participation rates in Nicaragua and Costa Rica [10,11].

The objective of this study is to examine the determinants of participating in two distinct afforestation programs in Taiwan. Specifically, we examine the probability of participating in afforestation programs on plain and sloped land using the multinomial logit model. We make several contributions to the literature on afforestation programs. Firstly, we examine the determinants of participating in two afforestation programs on plain and sloped land. The vast majority of studies only consider one policy or intervention [5,6,7,8]. This distinction is significant because afforestation programs on plain and sloped land will vary based on the economic incentives to enroll, the financial support provided to participants, and policy details. To the best of our knowledge, this paper is the first to explicitly consider the participation decision in two afforestation programs based on plain or sloped land. Secondly, we use large population-based data from agricultural censuses. Many findings in prior studies have sample sizes of just a few hundred participants in a single county or region, constraining the general applicability of these findings [8,12]. Finally, we conduct a simulation exercise to compare how the participation decision varies based on the forest farm operator’s socio-demographic characteristics. This analysis allows us to build on prior research examining how governments could specifically target groups with low participation rates to enroll in these programs.

### Background on the Forest Industry in Taiwan

Forests heavily dominate Taiwan’s environment and geography. Statistics from the Forestry Bureau of the Council of Agriculture show that forests comprise 21,970 of Taiwan’s 36,197 km^2^ land area [13]. However, deforestation has become increasingly prevalent due to economic development, urban expansion, and illegal timber trade. Current estimates show that Taiwan’s net loss of forest cover totaled 672 km^2^ from 2010 to 2015 [14].

Taiwan has enacted several afforestation programs to combat these losses. Since 1951, the government has provided subsidies to individuals who invest in ecological conservation under the “Detailed Rules and Regulations of Rewarding Afforestation of Ecological Conservation Land” [15] (López-Pujol, 2011). Initially, only forest farm households cultivating plots on plain land were eligible for these programs. However, additional measures were implemented to include forest farm households managing plots on sloped land due to the ecological destruction caused by Typhoon Herb in 1996 [15]. Sloped lands are areas with an elevation above 100 m or less than 100 m with an average slope of more than 5% [16].

Plain and sloped lands have different categorizations and limitations on their use. Plain land is classified into agricultural or pastoral lands, and farmers may use these areas to feed or plant in any manner, including afforestation. In contrast, sloped land is classified into agricultural or forest lands. If the sloped land is designated as forest land, then it can only be used to plant trees or beekeeping and growing mushrooms or clematises. Farmers can collect non-timber products from these lands as long as they comply with these conditions [17]. The most recent afforestation policy, known as the Green Forestation Plan, provides financial support to forest farmers that plant trees on plain and sloped land. The Green Forestation Plan was implemented in 2009 and has reforested more than 22,310 hectares of forestland, with an estimated initial cost of NTD 30 billion or USD 1 billion [17,18]. These two policies are subject to several conditions and their own respective rules and regulations. Firstly, afforestation must occur on agricultural and pastoral land in non-urban planning zones. Planting can also occur in aboriginal areas on sloped lands [19]. Secondly, the minimum planting area for these programs is 0.5 and 0.1 hectares for plain and sloped lands, respectively [19,20].

The Forestry Bureau manages the implementation and regulation of these two policies. On both of these lands, the government provides seedlings of approved tree varieties while ensuring that planted trees meet environmental standards and are evenly distributed. If these prerequisites are satisfied, then the government will monitor these forests and issue annual subsidies for 20 years. These payments range from NT 210,000 to 110,000 (US 7000 to 3667) per hectare, with tree survival rates from 70% to 40% on plain land [20]. In contrast, this amount is reduced to NT 120,000 to 20,000 (US 4000 to 667) per hectare and survival rates of at least 70% that decreases annually by 2% on sloped land [19]. Finally, executive agencies will take samples to monitor these forests and review all reports submitted by participating forest farm households.

## 2. Materials and Methods

### 2.1. Data

The data used in this study is drawn from the 2010 and 2015 Agricultural Census Survey in Taiwan [21]. The respondents in each wave of the census survey include all registered farm households in Taiwan. During an in-person interview, one principal farm operator reports details of their farm’s production and family characteristics. We used census data from 2010 and 2015 since these two surveys provide information regarding each household’s participation decision in afforestation programs. Due to privacy concerns, the census surveys with information on individual forest farm households are not publicly available. Readers interested in this dataset should send an application form to the Directorate-General of Budget, Accounting, and Statistics of the Executive Yuan.

We begin by limiting our sample to only forest farm households. In the Agricultural Census Survey, forest farms are defined as family farms that have at least 0.1 hectares of forestland. Additionally, this land must have been used in forest-related activities, such as forest production and agri-tourism in the survey year. Our samples include 83,009 and 87,030 forest farm households in 2010 and 2015, respectively. The dependent variable of interest is the participation decision of each farm household in afforestation programs. The participation decision of farm households in these two types of afforestation programs on plain or sloped land is documented in the surveys. Accordingly, we construct a choice variable reflecting the four possible outcomes of participation: zero for farms participating in neither program; one for farms participating in the program on plain land only; two for farms participating in the program on sloped land only; and three for farms participating in both of these programs.

Similar to the specifications of previous studies on afforestation programs, several explanatory variables associated with the socio-demographic characteristics of farm operators, farm production, and family farm characteristics are defined [22,23]. These variables have been shown to be associated with the participation decision in afforestation programs. For the socio-demographic characteristics of the farm operator, we create a continuous variable for age and a dummy variable for gender. We also construct five other dummy variables for educational attainment: illiterate, elementary, junior high, senior high, and college-educated or higher. With respect to family structures, we generate four variables indicating the number of household members and their age: males above/below 15 years old and females above/below 15 years old. Other variables reflect the production characteristics of each respective forest farm. We define a continuous variable to account for the size of forest land. Another variable is constructed for the ratio of the size of land owned by each household. Concerning labor allocation, we control for the number of hired workers and the ratio of household members working full-time on the self-owned forest farm.

Table 1 reports the sample statistics and detailed definitions of all variables. Summary statistics show that 25,494 forest farm households participated in at least one afforestation program. Among these participants, 7783, 17,165, and 546 households enrolled in afforestation programs on plain, sloped, or both lands. Participants and non-participants are similar in terms of age, gender, education, and household characteristics. However, forest farm households participating in any afforestation program have more hired workers compared to non-participants. Additionally, participants have larger plots of land compared to their counterparts. The average size of forest lands for non-participants is 1.369 hectares, compared to 1.766, 2.828, and 4.626 hectares for households participating in plain, sloped, and both afforestation programs.

### 2.2. Analytical Framework

The objective of our analysis is to estimate the effects of the determinants on forest farms’ participation in afforestation programs. We consider afforestation programs on both plain and sloped land in this study. Therefore, there are four possible outcomes: no planting at all, only planting on plain land, only planting on sloped land, and plating on both types of land. Following the literature on discrete choice models, the participation decision of forest farms in afforestation programs can be specified as the multinomial logit model (MNL). If we define a choice variable *j* whose value is zero for non-participants, one for farms that participate in afforestation on plain land, two for farms that participate in afforestation on sloped land, and three for farms that participate in both programs, the probability of each farmer’s decision is specified as follows [24]:(1)$${Pr}_{j}=\frac{exp(X’\beta_{j})}{\overset{4}{\underset{j=1}{\sum }}\exp(X’\beta_{j})},j=0,1,2,3$$
where Pr*_j_* is the probability of the farm participation in the $j^{th}$ type of afforestation program, and *X* is a vector of farm and household characteristics that are associated with the participation decision. $\beta_{j}$ are the parameters to be estimated. For the model’s identification purpose, we normalize $\beta_{0}$ to be zero (i.e., the non-participant group). Consistent estimates can be obtained by using the full-information maximum likelihood estimation method on the following log-likelihood function:(2)$$lnL=\overset{n}{\underset{i=1}{\sum }}ln({Pr}_{j})$$
where *j* = 0, 1, 2, 3.

We also cluster our estimates at the township level to generate more conservative results.

## 3. Results and Discussion

### 3.1. The Determinants of Participating in Afforestation Programs

Table 2 presents the estimation results from the multinomial logit model. For the model’s identification purpose, forest farm households that do not participate in any afforestation programs are the reference group. As reported in Table 2, the estimated coefficients of the explanatory variables vary across different types of afforestation programs. The value of the F-test under the null hypothesis that the estimated coefficients across different choices of afforestation programs are statistically equal is 833.97 (*p*-value < 0.001). This result shows that there is a need to distinguish between these two types of afforestation programs. In other words, using a restricted model with only a binary indicator for participation in afforestation programs could result in misleading findings. The details of the computer programs we used in the empirical estimation can be found in the online Appendix A.

To explore the magnitude of the effects of each explanatory variable on the choice to participate in afforestation, we calculate the marginal effects of the multinomial logit model and present these results in Table 3. The marginal effects explore the change in the likelihood of forest farms enrolling in each type of afforestation program in response to a one unit change in the explanatory variable [25].

The results indicate that forest farm operators with higher educational attainment are more likely to participate in afforestation programs. Compared to non-participants, forest farm operators with a college degree are more likely to participate in afforestation programs on plain and sloped lands by 1.34% and 2%, ceteris paribus. This finding is consistent with Claytor et al. [26] and Legesse et al. [6], reflecting the possibility that educated individuals are more likely to be attracted to the positive environmental and social externalities of these policies. Another explanation is that education allows forest farmers to gain the skills and knowledge to properly engage in afforestation [6].

Farm characteristics are also associated with the decision to enroll in afforestation programs. Each additional hired worker increases the probability of a household participating in plain, sloped, and both afforestation programs by 2.86%, 7.03%, and 0.22%. One explanation for this finding is that afforestation requires large amounts of initial labor and capital to begin plantations, even though forest management becomes less demanding over time [27]. Farming in Taiwan is characterized by small-scale family farm households, suggesting that many forest farms are unable to plant these trees without additional farm labor. Similarly, a one hectare increase in land increases the probability of participating in sloped land and both afforestation programs by 0.31% and 0.01%, ceteris paribus. These results are consistent with Duesberg et al. [28] and Powlen and Jones [11]. Larger forest farms can dedicate more land to afforestation while still having additional property available for other purposes, which is not the case for small forest farms. Increases in the ratio of self-owned farmland also raises the participation rate of plain, sloped, and both afforestation programs by 3.30%, 7.84%, and 0.39%, respectively. Similar to the findings of Nguyen et al. [5] and Legesse et al. [6], forest farm households that own more of their forest land are more likely to rehabilitate it with forests, since they become key legal players in managing their respective properties. These owners could also be more likely to view their forest land as an asset because of their ownership [6].

### 3.2. Simulation Results for Policy Implications

While Taiwan has been actively promoting afforestation programs, agricultural authorities should target groups with low participation rates to ensure that these programs are more effective. We perform a simple simulation exercise to demonstrate how the participation decision in each type of afforestation program varies by the age, education, and gender of the principal farm operator. The simulation analysis is conducted as follows. We change the values for the farm operator’s education level, age, and gender while fixing other variables at their sample means. These values are then used to calculate the probability of participating in each type of afforestation program. Thus, we can compare how the participation decision can vary based on their socio-demographic characteristics across programs.

Table 4 reports the results from the simulation exercise. We find that the probability of participation in afforestation on plain land increases as farm operators age, ranging from 2.99% to 5.21% depending on the level of educational attainment. Similar trends are observed for the probability of farm operators participating in afforestation on sloped land, ranging from 6.47% to 10.08%. Finally, the probability of participating in both afforestation programs is small at 0.17% to 0.29%.

These findings have several policy implications. Our results suggest that age is correlated with increased participation in afforestation, likely because older farmers are more experienced. To increase participation rates in afforestation, governments should establish additional training programs that clearly instruct younger farmers on how to plant and maintain these lands. One specific policy suggestion could involve the inclusion of organizations like agricultural cooperatives. In particular, Taiwan has been successful in encouraging these organizations to guide young farmers by providing basic technical training and partnering with private financial institutions to help these individuals obtain land [29]. We suggest that governments partner with agricultural cooperatives to enroll and train farmers from this specific sub-group to participate in afforestation programs. These initiatives would be useful since prior research has shown that information asymmetries are common when forest farms seek to enroll in these programs [30].

We also suggest that authorities must consider the type of land when implementing afforestation policies and the amount of financial support that is provided to participants. For example, since plain land can be used to produce economically lucrative crops, there are higher opportunity costs when farm operators use these lands for afforestation in Taiwan. While these afforestation programs already consider economic incentives, the Taiwanese government should expand the amount of this subsidy to increase participation in afforestation programs. In the case of the Green Forestation Plan, the program was expected to reforest up to 600 km^2^ of land [31]. However, the program planted less than half of that amount (223.10 km^2^) by 2018. Our results suggest that participation rates in afforestation programs on plain land are particularly low. This finding corroborates prior research showing that farm households in Taiwan believe that the subsidy amount is too meager [31]. Thus, the government can encourage farmers to enroll in afforestation programs by increasing the subsidy amount to participants planting on plain land.

## 4. Conclusions

A considerable body of literature has focused on afforestation programs, with the primary emphasis focusing on the association between farm and household characteristics and the participation decision of forest farms. As policymakers seek to increase the coverage rate of afforestation programs, it is crucial to have a better understanding of forest farms’ behavior towards these measures. Previous studies have solely considered one category of afforestation program. This study fills the knowledge gap by considering two different types of afforestation programs. We use a population-based sample of forest farms drawn from agricultural census surveys in Taiwan. Our utilization of large-scale data allows us to ease some of the errors caused by random sampling.

Several interesting findings are revealed. Socio-demographic characteristics such as age and educational attainment increase the probability of enrolling in afforestation programs. Additionally, farm production characteristics such as the ratio of household members working on the self-owned farm, the number of hired workers, the size of forest land, and the ratio of self-owned land also increase participation rates. Our results also suggest that policymakers must consider the importance of land type when enacting afforestation programs.

Although this paper presents some interesting results, some caveats exist. Firstly, environmental quality is likely to be associated with forest farms’ participation in afforestation programs. However, it is empirically challenging to specify these types of variables in econometric analyses. If the geographic location of each forest farm or forestland can be identified, further studies can use the Geographic Information Systems (GIS) method to construct some indicators of environmental characteristics, such as soil quality and land productivity. This technique would check the robustness of our findings. Secondly, it would be interesting to see how participation in afforestation programs is associated with the wellbeing of forest farmers. However, this analysis would require detailed information on forest farm incomes. Collecting data on forest farm incomes is difficult since this information is sensitive due to confidentiality concerns. Thirdly, our estimates may suffer from omitted variable bias since we cannot control for all characteristics that are associated with forest farms’ participation decisions (e.g., non-market values or the personality traits of the farm operator). Finally, the robustness of our findings can also be checked by using panel data that further controls for the time-invariant unobserved factors affecting forest farm households. To the best of our knowledge, it is even more difficult to construct a panel dataset of forest farms using a population-based census survey.

## Figures and Tables

**Table 1 ijerph-17-03962-t001:** Sample statistics of the selected variables.

Variable	Definition	All Sample		None		Afforestation _Plain Land		Afforestation _Slope Land		Both Programs	
		Mean	S.D	Mean	S.D	Mean	S.D	Mean	S.D	Mean	S.D
MNL	An indicator of participation in afforestation programs (0–3).	0.257	0.642	0	0	1	0	2	0	3	0
Age	Age of the farm operator (year).	50.930	15.893	50.869	16.072	50.763	15.872	51.508	14.316	51.357	15.219
Male	If the farm operator is male (=1).	0.785	0.411	0.790	0.407	0.781	0.413	0.744	0.437	0.725	0.447
Illiterate	If the farm operator is illiterate (=1).	0.074	0.261	0.077	0.266	0.069	0.253	0.051	0.221	0.068	0.252
Elementary	If the farm operator finished elementary school (=1).	0.364	0.481	0.366	0.482	0.309	0.462	0.363	0.481	0.383	0.487
Junior high	If the farm operator finished junior high school (=1).	0.219	0.414	0.222	0.416	0.183	0.387	0.215	0.411	0.187	0.390
Senior high	If the farm operator finished senior high school (=1).	0.235	0.424	0.233	0.423	0.254	0.435	0.244	0.429	0.205	0.404
College	If the farm operator has college education (=1).	0.108	0.311	0.102	0.302	0.185	0.388	0.127	0.333	0.158	0.365
Child_male	Number of male household member age <15 (person).	0.187	0.525	0.182	0.515	0.163	0.499	0.248	0.614	0.170	0.475
Child_female	Number of female household member age <15 (person).	0.169	0.507	0.163	0.499	0.148	0.475	0.223	0.576	0.201	0.578
Adult_male	Number of male household member age ≥15 (person).	1.611	1.026	1.612	1.024	1.534	0.978	1.641	1.062	1.615	1.042
Adult_female	Number of female household member age ≥15 (person).	1.368	1.053	1.361	1.052	1.328	1.021	1.436	1.075	1.447	1.055
R_selffarm	Ratio of household members working full-time on self-farm.	0.042	0.162	0.033	0.144	0.111	0.258	0.085	0.220	0.117	0.256
Hired workers	Number of hired workers (person).	0.683	1.136	0.507	0.915	1.408	1.194	1.802	1.792	1.870	1.550
Land	Size of the forest land (hectare).	1.545	5.261	1.369	4.287	1.766	12.397	2.828	6.434	4.626	13.849
R_self own land	Ratio of self-own land.	0.773	0.412	0.758	0.423	0.862	0.332	0.862	0.338	0.900	0.265
Year 2015	If in year 2015 survey (=1).	0.512	0.500	0.512	0.500	0.542	0.498	0.496	0.500	0.465	0.499
Number of farms		170,039		144,545		7783		17,165		546	

Data was drawn from the 2010 and 2015 Agricultural Census Surveys in Taiwan.

**Table 2 ijerph-17-03962-t002:** Estimated coefficients of the multinomial logit model.

Variable	Afforestation _Plain Land			Afforestation _Slope Land			Both Programs		
	Coefficient		S.E	Coefficient		S.E	Coefficient		S.E
Age	0.002	*	0.001	0.003	***	0.001	−0.002		0.003
Male	−0.014		0.032	−0.283	***	0.023	−0.306	***	0.106
Elementary	−0.184	***	0.051	0.272	***	0.041	0.047		0.181
Junior high	−0.223	***	0.054	0.244	***	0.043	−0.161		0.197
Senior high	−0.038		0.053	0.222	***	0.042	−0.224		0.195
College	0.384	***	0.055	0.304	***	0.046	0.234		0.203
Child_male	−0.060	**	0.027	0.110	***	0.017	−0.171	*	0.096
Child_female	−0.026		0.028	0.114	***	0.017	0.139	*	0.084
Adult_male	−0.109	***	0.014	−0.047	***	0.010	−0.047		0.047
Adult_female	−0.063	***	0.013	−0.079	***	0.010	−0.035		0.045
R_selffarm	0.775	***	0.051	0.222	***	0.045	0.737	***	0.170
Hired workers	0.899	***	0.010	1.021	***	0.009	1.028	***	0.019
Land	0.014	***	0.004	0.043	***	0.002	0.046	***	0.003
R_self own land	1.034	***	0.036	1.148	***	0.027	1.603	***	0.150
Year 2015	0.185	***	0.029	0.098	***	0.022	−0.119		0.107
Constant	−4.438	***	0.085	−4.219	***	0.066	−7.489	***	0.318

Note: ***, **, * indicates significance at the 1%, 5%, and 10% level, respectively.

**Table 3 ijerph-17-03962-t003:** Estimated marginal effects of the multinomial logit model.

Variable	Afforestation _Plain Land			Afforestation _Slope Land			Both Programs		
	Mar. Eff		S.E	Mar. Eff		S.E	Mar. Eff		S.E
Age	0.00%		0.000	0.03%	***	0.000	0.00%		0.000
Male	0.19%		0.001	−2.15%	***	0.002	−0.07%	**	0.000
Elementary	−1.00%	***	0.002	2.25%	***	0.003	0.00%		0.001
Junior high	−1.14%	***	0.002	2.09%	***	0.003	−0.06%		0.001
Senior high	−0.34%		0.002	1.76%	***	0.003	−0.09%		0.001
College	1.34%	***	0.002	2.00%	***	0.004	0.04%		0.001
Child_male	−0.34%	***	0.001	0.91%	***	0.001	−0.06%	**	0.000
Child_female	−0.21%	*	0.001	0.89%	***	0.001	0.04%		0.000
Adult_male	−0.42%	***	0.001	−0.27%	***	0.001	−0.01%		0.000
Adult_female	−0.20%	***	0.001	−0.55%	***	0.001	0.00%		0.000
R_selffarm	3.04%	***	0.002	0.99%	***	0.003	0.19%	***	0.001
Hired workers	2.86%	***	0.000	7.03%	***	0.001	0.22%	***	0.000
Land	0.02%		0.000	0.31%	***	0.000	0.01%	***	0.000
R_self own land	3.30%	***	0.002	7.84%	***	0.002	0.39%	***	0.001
Year 2015	0.69%	***	0.001	0.61%	***	0.002	−0.05%		0.000

Note: ***, **, * indicates significance at the 1%, 5%, and 10% level, respectively.

**Table 4 ijerph-17-03962-t004:** Simulation results of participation in afforestation programs.

Age	Elementary	Junior High	Senior High	College	Female	Male
Participate in the Plain Land Afforestation Program Only						
20	3.16%	2.99%	3.45%	4.88%	3.56%	3.59%
30	3.21%	3.03%	3.49%	4.94%	3.61%	3.65%
40	3.25%	3.07%	3.54%	5.01%	3.66%	3.70%
50	3.30%	3.12%	3.59%	5.08%	3.71%	3.75%
60	3.34%	3.16%	3.64%	5.14%	3.76%	3.81%
70	3.39%	3.20%	3.69%	5.21%	3.81%	3.86%
**Participate in the Slope Land Afforestation Program Only**					
20	8.07%	8.22%	8.02%	8.66%	8.40%	6.47%
30	8.32%	8.47%	8.27%	8.93%	8.67%	6.68%
40	8.58%	8.74%	8.53%	9.21%	8.94%	6.89%
50	8.85%	9.01%	8.80%	9.49%	9.21%	7.11%
60	9.13%	9.29%	9.07%	9.78%	9.50%	7.33%
70	9.41%	9.58%	9.35%	10.08%	9.79%	7.57%
**Participate in Both Programs**					
20	0.24%	0.20%	0.19%	0.27%	0.29%	0.22%
30	0.23%	0.20%	0.19%	0.27%	0.28%	0.21%
40	0.23%	0.19%	0.18%	0.26%	0.28%	0.21%
50	0.22%	0.19%	0.18%	0.26%	0.27%	0.20%
60	0.22%	0.19%	0.18%	0.25%	0.26%	0.20%
70	0.21%	0.18%	0.17%	0.25%	0.26%	0.20%

Note: All other variables are fixed at their sample means.

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
