# Peer review of "What Determines Forest Farmers’ Participation in Afforestation Programs? Empirical Evidence from a Population-Based Census Survey"

_ijerph, 2020, doi:10.3390/ijerph17113962_

Round 1
Reviewer 1 Report
the paper analyzes the participation of farmers in forestation campaigns in Taiwan, highlighting the impact of the various socio-demographic and environmental variables in the decision to participate in the programs.
Objectives, methodology and future research perspectives are clear and well exposed.
I want to suggest two elements of reflection that could be included in the study:
- Can the economic dimension of the company be taken into account in the model? or farmers' income? how does this variable influence the probability of participation?
- in terms of public policy, what is the cost of these programs? what are the current and future environmental benefits (e.g. value of the carbon stock capacity of new forests?).
Reviewer 2 Report
Lines 30-33: This sentence has three “such as”
Lines 49-52: Why results in the introduction?
Lines 55-56: The vast majority? You only mention two studies.
Line 59: considers
Line 69: Why different units?
Line 117: account for?
Table 1 and others: Don´t see the need for this character: “_”
After line 132, numbering begins again. The comments below follow this new numbering:
Line 44: Is it really an increase from year to year?
Lines 46-47: Is this less than 1%. Is this result really important?
Line 64 and subsequent: This is methodology, not results
Results and discussion: It is mostly results. Very few references are discussed
Reviewer 3 Report
The study use agricultural census data of 2010 and 2015 gathered in Taiwan to estimate the determinants of the participation of forest farm households in afforestration programs. The subject is worth investigating and the data set has some merits. However, the paper is not well connected to the literature, omit relevant information and does not properly interpret the results. Recommendations are also not based on results. Thus major revisions are necessary in order to make the paper publishable.
General issues
1) The literature review in the introduction should be improved and extended. Currently it is insufficient. Among others the following literature should be taken into account.
Powlen, K. A., & Jones, K. W. (2019). Identifying the determinants of and barriers to landowner participation in reforestation in Costa Rica. Land use policy, 84, 216-225.
Baker, K., Baylis, K., Bull, G. Q., & Barichello, R. (2019). Are non-market values important to smallholders' afforestation decisions? A psychometric segmentation and its implications for afforestation programs. Forest policy and economics, 100, 1-13.
Legesse, B. A., Jefferson-Moore, K., & Thomas, T. (2018). Impacts of land tenure and property rights on reforestation intervention in Ethiopia. Land Use Policy, 70, 494-499.
Aganyira, K., Kabumbuli, R., Muwanika, V. B., Tabuti, J. R., & Sheil, D. (2020). Determinants of participation in state and private PES projects in Uganda. Scientific African, e00370.
Nielsen, A. S. E., Jacobsen, J. B., & Strange, N. (2018). Landowner participation in forest conservation programs: A revealed approach using register, spatial and contract data. Journal of Forest Economics, 30, 1-12.
2) The research is restricted to “forest farm households”. However, it is not clear how these household are defined. Please clearly describe what delineates a forest farm household from a farm household. Further, please characterized these farms in order to enable the reader to understand the farming system. According to the data in Table 1, only a very small number of family members work on the farm (only 4,2% or family members work on the farm, on average). Actually, it is hard to believe this number. It seems that most of the farms mainly rely on hired workers. Are these salaried full-time or part-time workers? You need provide more information on the forest farming systems.
3) The attributes of the different programs are explained on page 2. However, you should provide more information. Since when is the “Green Forestation Plan” in place? Can farms use the afforest areas? Can they collect NTFP? Can you provide also some information on how many ha have been afforested in the different programs since its establishment? Was there any change in the program between 2010 and 2015? Please provide more information on the programs.
4) Policy recommendation should be based on results! Your recommendations are either not based on results or they are trivial. You write “To increase participation rates in afforestation, governments should establish additional training programs that clearly instruct farmers on how to plant and maintain these lands.” (line 82-82), however, you did not examine the effect of training programs, but you speculate that age is related to experience. But how this is related to training programs is by far not clear. You also write: “Second, we suggest that authorities must also consider the type of land when implementing afforestation policies and the amount of financial support that is provided to participants. For example, since plain land can be used to produce economically lucrative crops, there are greater opportunity costs when farm operators use these lands for afforestation in Taiwan. Thus, economic incentives must be considered when designing and enacting afforestation programs on plain, sloped, and other types of land. (line 86-91). I think this is a rather trivial recommendation and also not based on your results! The afforestration program in Taiwan already fully take this into account. To be honest, I doubt that your results can lead to any relevant policy recommendations since none of the factors that you investigated can be directly influenced by policy! Does policy what to increase the number of hired workers, or the number of boys on the farm in order to increase participation in the slope land program? All relevant design variables of the afforestration programs (minimum amount of land, compensation payments, and others) which could be influenced by policy were not investigated.
5) Your analysis omitted a lot of relevant factors affecting the participation decision (for example training, opportunity costs, slope of the land managed by the farm). However, this omission (and limitation) is nowhere mentioned or discussed.
Specific issues
Page 1, Abstract: “We find that failing to distinguish between afforestation programs may result in misleading findings.” I don’t think that this can be concluded from your research. You need to do additional calculations to support this claim.
Page 1, Abstract: “are more likely to enroll in afforestation programs on sloped land due to lower opportunity costs.” I don’t think that this conclusion is supported by your result. You don’t have any direct information on opportunity costs. Please clarify.
Page 1,first para: „…directly paying participants to retire or plant tree cover on land…” This reads quite strange “paying participants to retire or plant tree”. Please clarify.
Page 1,first para: ”Thus, many countries such as China…” If many countries implemented afforestration policies you should not provide just one example. Provide more.
Page 1, second para: “…many of these studies only consider one homogeneous policy that does not account for different types of land.” If this is the case for “many of these studies” then there should be some studies available which did consider heterogeneous policies. Please clarify and name these studies. What do you mean by “types of land”?
Page 2, first para: “Other studies have considered landowners’ responses towards afforestation programs in Ethiopia, Vietnam, and the United States (Tadesse et al., 2017; Nguyen et al., 2010; Kim and Langpap, 2016).” What were the findings of these studies? There are more relevant studies that need to be considered.
Page 2, second para: “Farm characteristics such as the number of hired workers and whether households owned their own land are also correlated with participation.” Yes, correlation is not causation. This must be better reflected when presenting and discussion the results.
Page 2, second para 2: “Finally, we conduct a simulation exercise to provide insight on how policymakers can encourage farms with low adoption rates to enroll in these programs.” The simulation exercise is not related to possibilities of policy makers. Policy maker may change the features of the programs, however this is not investigated at all. Please correct.
Page 2, third para: “The vast majority of prior studies fails to differentiate between land types (Nguyen et al., 2010; Duesberg et al., 2013).” When only the vast majority fails to differentiate between land types, then there should be some studies which did not fail. Please cite them and refer to their findings. You do also not investigate the land type! You did not include any information on the characteristics of the land managed and owned by the households (for example its slope).
Page 2, fourth para: “forests comprise of 2,197,000 hectares on Taiwan’s 36,197 km2 land area…” Please express the forest land also in km2!
Page 3, para 3: “Similar to the specifications of previous studies on afforestation programs (Ryan et al., 2018)..” The similarities and differences should be more reflected. There should also be some sound reasoning why variables are included. There should be a reflection on omitted variables.
Table 1: “MNL” might be better named “Participation”. The definition of R_selffarm is unclear. Is it really the case that only 4,2% of the household members work on the farm? How is the ratio calculated? Same holds for R_self own land. How is the ration calculated.
Line 5-6: “Following the conventional wisdom from the literature on discrete choice models,..” What do you want to say by “following the conventional wisdom”? Clarify or reformulate.
L 27-28: “In other words, failure to differentiate between afforestation programs could result in misleading findings.” You could test for this claim if you just run an logit model on participation in the program regardless of the type or program. So far, you did not test this claim.
L 44-47: “Our findings show that increases in the number of children and adults lowered participation rates in the afforestation program on plain land by 0.20 to 0.42 percentage points. However, farm households with more boys and girls are 0.91 and 0.89 percentage points more likely to plant on sloped land.” Do you think that there is any meaningful explanation for this result. Is this really relevant? I don’t think so.
Line 100: „Our utilization of large-scale data allows us to ease some of the errors caused by random sampling.” Random sampling can be a very effective tool and can gather all relevant data for the specific purpose. The utilization of large-scale date comes at the costs of being constraint to the data that were collected for other purposes. You omit a lot of relevant data. This omission must be acknowledged as a main limitation.
Round 2
Reviewer 3 Report
Thank you for taking all my remarks carefully into account. The revisions are to my satisfaction. I just would like to request you to make an explicit statement about the data availability, the data format and the statistical software you used to make your estimations. Of course it would be highly appreciated if you would make your data set fully available to the readers.
